

# Robust coffee plant disease classification using deep learning and advanced feature engineering techniques

Hanin Ardah[1], Maher Alrahhal[2], Walaa M. Abd-Elhafiez[3,4] and Doaa Trabay[5]

[1] Department of Computer Sciences, College of Computer and Information Sciences, Princess Nourah bint Abdulrahman University, Riyadh, Saudi Arabia
[2] Computer Science and Engineering Department, JNTUH University College of Engineering, Science & Technology Hyderabad, Hyderabad, Telangana, India
[3] Computer Science Department, Faculty of Computers and Artificial Intelligence, Sohag University, Sohag, Egypt
[4] College of Engineer and Computer Science, Jazan University, Jazan, Saudi Arabia
[5] Information System Department, Obour Institutes for Obour High Institute for Management and Computers and Information Systems, Cairo, Egypt

Corresponding authors
Maher Alrahhal,
maherrahal92@gmail.com
Walaa M. Abd-Elhafiez,
w_a_led@yahoo.com

## ABSTRACT

Coffee, the world's most traded tropical crop, is vital to the economies of many producing countries. However, coffee leaf diseases pose a serious threat to coffee quality and sustainable production. Deep learning has shown strong performance in plant disease identification through automatic image classification. Nevertheless, reliance on a single convolutional neural networks (CNNs) architecture restricts feature variability and real-world generalization. Moreover, limited work has systematically combined feature selection/reduction with CNNs, which constrains the advancement of hybrid models capable of capturing complementary features while ensuring computational efficiency without accuracy loss. This article presents an enhanced deep learning-based framework for coffee disease classification incorporating a hybrid strategy that integrates CNNs and advanced feature selection algorithms. GoogLeNet and ResNet18 are paired for complementary feature extraction, Principal Component Analysis (PCA) and Singular Value Decomposition (SVD) are employed for dimensionality reduction, and ANOVA and Chi-square are applied to select the most informative features. An Adam optimizer (learning rate = 0.001, batch size = 20, epochs = 50) with early stopping is used for training. Experiments on the BRACOL dataset achieved 99.78% accuracy, with precision, recall, and F1-score all exceeding 99% across classes. To the best of our knowledge, this study systematically integrates GoogLeNet and ResNet18 with PCA/SVD dimensionality reduction and analysis of variance (ANOVA)/Chi-square feature selection, for coffee disease classification, thereby addressing a key gap in prior research.

## INTRODUCTION

The world's food security depends on agriculture, which includes the production of crops and livestock. However, this sector still faces challenges because plant diseases are a major threat to agricultural productivity. In addition to decreasing crop yields and jeopardizing the rustling of produce, these diseases damage the economic foundations of farming communities. Coffee plant diseases, particularly coffee leaf rust (*Department of Primary Industries and Regional Development, 2021*), have become a major concern among these challenges due to their substantial impact on coffee production.

One of the most widely consumed beverages worldwide and a significant agricultural product, coffee contributes significantly to both the national and international economies (*Lu, Tan & Jiang, 2021*). Coffee leaf rust is a destructive fungal disease that primarily affects Arabica coffee and has historically caused large financial losses (*Kongsom & Panyakul, 2016*; *Ditlevsen, Sandøe & Lassen, 2019*). Coffee leaf rust was first discovered in the 19th century and used in Arabica coffee plantations to be replaced with tea, highlighting the significant influence of these diseases (*Oliveira et al., 2014*).

Despite advancements in creating rust coffee varieties and enhancements in technology, coffee leaf rust continues to lead to production decreases of 20% or higher in affected regions (*ICO, 2021*; *Mokhtar et al., 2015*; *Hossain, Hossain & Rahaman, 2019*; *Senanayake & Maduranga, 2023*). Traditional disease management methods, such as crop rotation, chemical treatments, and the use of resistant cultivars, offer few and often reactive solutions. There is an urgent need for more proactive and effective disease management strategies, given the rising demand for coffee globally.

Recent advancements in Artificial Intelligence (AI), particularly in machine learning (ML) and deep learning, offer promising improvements to current strategies for early disease identification and management (*Binney & Ren, 2022*). Deep learning has achieved great progress for plant disease detection in recent years, allowing automatic and very efficient classification of crop images. Most existing approaches, however, adhere to a single convolutional neural network (CNN) architecture, which limits the variability of the extracted features and can undermine generalizability to real-world scenarios (*de Oliveira Aparecido et al., 2020*). Furthermore, while some work employ feature selection or dimensionality reduction, very few integrate these processes with deep CNN architectures in a systematic, optimized fashion. This leaves a gap in developing hybrid models that can capture complementary feature representations while mitigating computational complexity without sacrificing accuracy.

To address these limitations, we propose a combination strategy of GoogleNet and ResNet18 for joint feature extraction with combined GoogleNet and ResNet18 features, and Principal Component Analysis (PCA) and Singular Value Decomposition (SVD) for effective dimensionality reduction. Our contributions are:

- New CNN combination approach to a more expressive and richer feature description.
- Iteratively adjusted PCA and SVD parameters to provide the best accuracy-efficiency trade-off. The integrated features were processed using ANOVA and chi-square to identify redundant attributes and retain only the most informative ones.

- Extensive testing of the proposed framework on different metrics (accuracy, precision, recall, F1-score) and statistical testing for significance to validate superiority over baseline models.
- Testing generalizability through a fixed train-test split was used, and assessment on real-world data beyond the BRACOL dataset.

The feature extraction techniques for feature reduction are first optimized for the classification process, and then the performances of PCA and SVD are compared. ANOVA and chi-square tests for feature selection are then performed. The purpose of this study is to gain a better understanding of these aspects to develop more robust and accurate disease classification models. The importance of this research is to apply these state-of-the-art techniques to improve the state of the art in coffee disease detection and thereby enhance disease management and support the stability of coffee economies.

## MATERIALS AND METHODS
*Computing infrastructure:*
Operating System: Nova Gaming GPU-Windows 11
Hardware Specifications: Intel Core i7, 16 GB RAM
Programming Language: Matlab 2023b

### Dataset and availability
We utilized the BRACOL dataset, a publicly available dataset of coffee leaf images. The dataset can be accessed at the following DOI: (https://data.mendeley.com/datasets/c5yvn32dzg/2).

### Implementation and code availability
All implementation details, including preprocessing scripts, model training, and evaluation codes, are available in the following GitHub repository: (https://github.com/DrMaherAlrahhal/coffe-code). The repository includes a README file providing step-by-step instructions for reproducing the experiments.

The feature selection and dimensionality reduction were first performed using PCA and SVD, with a 99% variance retention threshold. This threshold was determined through iterative experiments to ensure an optimal balance between classification accuracy and computational efficiency. Following the integrated features were processed using ANOVA and chi-square to prune redundant attributes and retain only the most informative ones.

## RELATED WORK
The following categories apply to the *corpus* of research related to coffee disease diagnosis using state-of-the-art computational methodologies and ML:

Integration of big data and advanced technologies in agriculture: *Kittichotsatsawat, Jangkrajarng & Tippayawong (2021)* discussed the use of big data and advanced technologies in agriculture, highlighting the substantial deficiency in the application of big data to coffee cultivation. Though big data is expected to bring about a revolution in agriculture, it is not being adequately applied to coffee cultivation. This is indicative of

further effort being required for the adoption of such technologies in agriculture for the enhancement of efficiency and supply chain management.

Computational techniques for coffee disease detection: *Sorte et al. (2019)* created a computer-based system that can detect coffee leaf diseases with 95.24% accuracy. Their research demonstrates the efficacy of computational tools in early disease detection and management, emphasizing their potential to boost coffee crop productivity and quality.

ML algorithms for disease classification: *Esgario, Krohling & Ventura (2020)* confirmed the value of deep convolutional networks in addressing challenging classification problems by using the ResNet50 architecture for coffee leaf disease classification. By applying machine learning techniques and combining ML with correlation analysis of infection rates with weather patterns, *de Oliveira Aparecido et al. (2020)* came up with a solution to the disease and pest problems in agriculture. Additionally, *Novtahaning, Shah & Kang (2022)* used an ensemble approach to combine multiple models, such as VGG-16, ResNet-152, and EfficientNet-B0, with an accuracy of 97.31%. This demonstrates how ensemble approaches can greatly improve disease detection performance.

Feature engineering methods: *Esgario, Krohling & Ventura (2020)* also contrasted with the VGG-16 and ResNet50 models to predict the severity of the disease and biotic stress in coffee, and they recorded 95.47% and 95.63% accuracy, respectively. *Binney & Ren (2022)* also showcased the advantages of transfer learning utilizing ResNet-121 and VGG19 with 95.44% and 99.36% accuracy, respectively. The studies identify the huge effect of transfer learning and different CNN models on accuracy for classification. *Jepkoech et al. (2021)* proposed datasets titled Influence that used Arabica coffee leaves. A Deep CNN was used by *Lisboa, Lima & Queiroz (2021)* to identify and recognize coffee leaf disease and determine the severity of the disease.

Hybrid approaches and practical uses: *Ayikpa et al. (2022)* achieved an accuracy of 97.37% by combining deep learning and traditional machine learning techniques using the JMuBEN dataset. The best aspects of both methods are combined in this hybrid strategy to improve classification performance. However, *Kumar, Gupta & Madhav (2020)* presented a CNN model based on transfer learning to reduce training time.

*Yamashita & Leite (2023)* created a low-cost microcontroller board with 98% and 94% accuracy for on-site disease classification. Their work highlights the useful applications of cutting-edge technology in agriculture and emphasizes the significance of real-time, reasonably priced solutions for farmers.

Nevertheless, *Yebasse et al. (2021)* introduced a novel method that questions the "black box" characteristics typical of disease recognition systems in traditional approaches by integrating visualization techniques into disease identification methods. *Chowdhury (2021)* used an extensive preprocessing methodology in their study, where dimensions and filtration were part of the main steps. It was aimed at image preprocessing; the pre-processed images were used to extract the main feature function, and this extraction function was retrained with ML algorithms. Also, *Deng (2020)* classified the coffee leaf images into multiple classes, namely Healthy, four levels of rust, and red spider mite infestation in their research. The preliminary test included trying different neural networks. Accuracy rates of only 40.3% and 59.4% for the fully connected neural network

**Table 1 Summary of articles presented in related work.**

| Study | Methodology | Advantages | Disadvantages |
|---|---|---|---|
| *Binney & Ren (2022)* | Transfer learning with ResNet-121 and VGG19 architectures | Achieves high accuracies (95.44% and 99.36%); demonstrates the effectiveness of transfer learning | Transfer learning effectiveness may vary depending on the dataset and model parameters |
| *Kittichotsatsawat, Jangkrajarng & Tippayawong (2021)* | Integration of big data and advanced technologies in agriculture | Highlights the potential of big data for optimizing agricultural practices | Limited to specific datasets; scalability and real-world applicability not fully explored. |
| *Sorte et al. (2019)* | Computational system for detecting coffee leaf diseases | High accuracy (95.24%) in disease identification demonstrates potential for early disease detection | Limited to specific datasets; scalability and real-world applicability not fully explored |
| *Esgario, Krohling & Ventura (2020)* | ResNet50 architecture for coffee leaf disease classification | Effective in handling complex classification tasks; demonstrates the capability of deep CNNs | Focus on a single model; lacks exploration of alternative architectures or hybrid approaches |
| *Novtahaning, Shah & Kang (2022)* | Ensemble approach combining EfficientNet-B0, ResNet-152, and VGG-16 models | Achieves high accuracy (97.31%); demonstrates benefits of model combination | Increased computational complexity may be less efficient for real-time applications |
| *Esgario, Krohling & Ventura (2020)* | Comparison of VGG-16 and ResNet50 for disease severity classification | High accuracy (95.47% and 95.63%); shows the versatility of deep learning architectures | Focuses only on classification accuracy; lacks real-world implementation considerations |
| *Ayikpa et al. (2022)* | Hybrid approach combining traditional ML and DL techniques using the JMuBEN dataset | High accuracy (97.37%); leverages strengths of both traditional ML and deep learning | The complexity of hybrid models may hinder practical implementation; may require extensive computational resources |
| *Yamashita & Leite (2023)* | Low-cost microcontroller board for on-site disease classification using BRACOL and LiCoLe datasets | Real-time, affordable solutions for farmers; practical for on-site disease detection | Limited to specific datasets; may not be generalizable across different environments and crop diseases |

and transferred CNN, which was inspired by the AlexNet model, were obtained, respectively. Table 1 provides a summary of the articles on related work presented.

The reviewed studies highlight several important gaps and opportunities for further research:

- Big data integration: To fully harness the transformative potential of big data technologies in coffee cultivation, there is a pressing need to improve their integration.
- Model practicality: While deep learning and computational models show high accuracy, additional research is necessary to understand their practical application in real agricultural environments.
- Hybrid approaches: Combining deep learning and conventional machine learning methods can improve performance, but striking the correct balance between interpretability and model complexity remains difficult.
- Real-time solutions: For efficient disease management, especially in environments with limited resources, affordable, real-time classification tools are essential. This study addresses these research gaps by examining how pre-trained CNN architectures, feature reduction tactics, and feature selection techniques impact coffee disease classification. Our research seeks to bridge the gap between state-of-the-art machine learning models and their practical application in agriculture by developing feature extraction and selection techniques.

The proposed method was evaluated using a train-test technique with a 70–30 split. Performance metrics, including accuracy, precision, recall, F1-score, and receiver operating characteristic area under the curve (ROC-AUC), were used to assess model performance.

### Assessment metrics

To evaluate the performance of our model, we employ the standard confusion matrix technique, which serves as the foundation for several key performance metrics (*Alrahhal & Supreethi, 2020*; *Alrahhal & KP, 2021a*; *Rawia et al., 2023*; *Alrahhal & KP, 2021b*), as shown below.

Accuracy (Ac): Proportion of correctly classified samples.

$$Ac = \frac{TP + TN}{TP + TN + FP + FN}. \tag{1}$$

Error rate (ER): The proportion of incorrectly classified instances.

$$Er = \frac{FP + FN}{TP + FP + FN + TN}. \tag{2}$$

Precision (P): Proportion of true positives among predicted positives.

$$Pr = \frac{TP}{TP + FP}. \tag{3}$$

Recall (R): Proportion of true positives among actual positives.

$$Re = \frac{TP}{TP + FN}. \tag{4}$$

Specificity (SP): Proportion of true negatives among actual negatives.

$$Sp = \frac{TP}{TN + FP}. \tag{5}$$

Area under the curve (AUC): Performance of a binary classifier, measured by the area under the ROC curve. AUC value ranges from 0 to 1, indicating perfect classification.

F Measure (FM): Harmonic mean precision and recall.

$$\text{F-Measure} = 2 \times \left( \frac{\text{precision} \times \text{recall}}{\text{precision} + \text{recall}} \right). \tag{6}$$

Training time : Duration is required to train the model on the dataset.
Testing time : Duration required to evaluate the model on a new dataset.

## METHODOLOGY

This article provides an integrated methodology for building and evaluating a classification model by combining pre-trained CNNs with feature extraction, feature fusion, feature selection, dimension reduction, and SVM classification, as shown in Fig. 1. The following are the sequential steps in the proposed methodology:

1. Dataset preparation: The images are resized to a standard size, which ensures uniformity in the dataset and fits the model's input requirements. The pixel values are
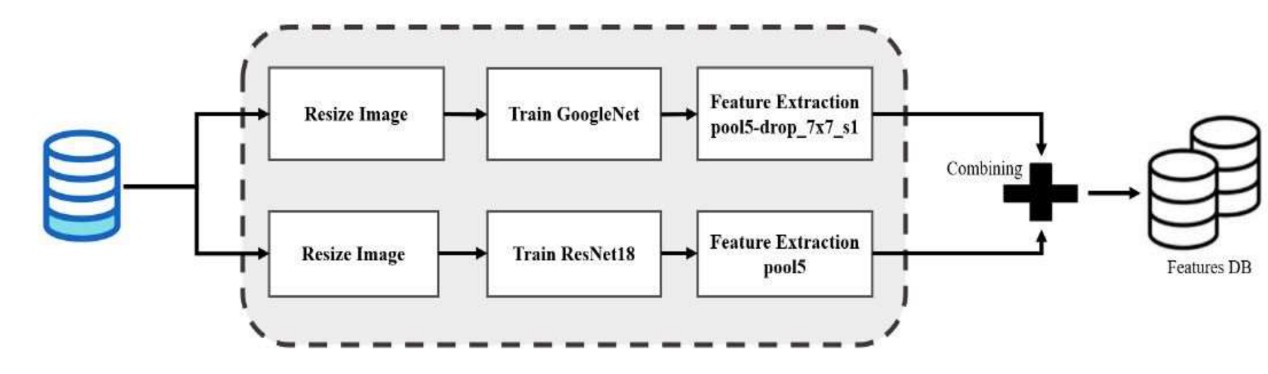

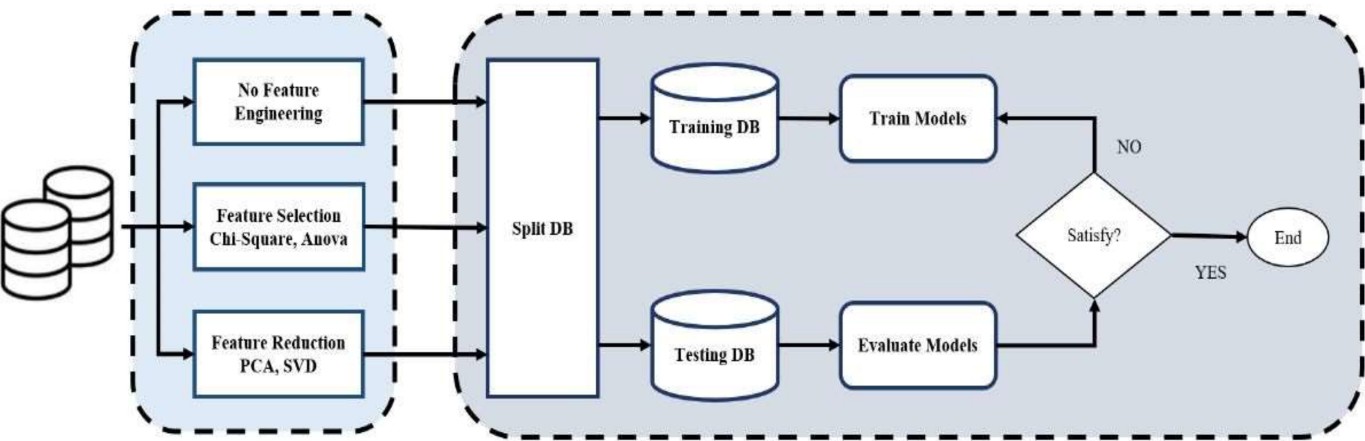

**Figure 1  Flowchart of proposed model.**                

then normalized, typically by scaling them to a particular range, to improve computational efficiency and model performance. The images can then be converted to grayscale, reducing the complexity of the data without the loss of essential structural and textural features. Images were resized to meet the input requirements of both ResNet18 and GoogleNet. Uniform preprocessing was applied across all models.

2. Transfer learning implementation: The final layers of the pre-trained CNNs were substituted to match the number of target classes to enable transfer learning from big datasets to the target classification task.

3. Training parameters: 70% of the data set is used for training and 30% for testing. The models are enhanced in terms of classification performance and generalization to new data by optimizing parameters on the training set. The model was trained on the model with the Adam optimizer, 0.001 learning rate, batch size of 20, and trained for 50 epochs. Early stopping when validation accuracy was not getting better for 10 consecutive epochs, and learning rate decay of 0.1 after 15 non-improving epochs.

4. CNN combination rationale: GoogleNet learns multi-scale spatial features through inception modules, while ResNet18 learns deep hierarchical features with residual

connections to avoid vanishing gradients. Complementarity arises from blending them and boosts robustness and accuracy.

5. Feature extraction and fusion: Features were extracted from the fourth-to-last layer of each CNN. The features were fused to form an integrated feature set that represents a very broad range of visual features. In this integration, multiple feature representations are learned to increase the overall robustness and performance of the classification engine.

6. Feature selection: The integrated features were subjected to ANOVA and chi-square pruning to retain the most informative features and eliminate redundancy.

7. Dimensionality reduction: PCA and SVD were employed with a 99% variance retention level, which was decided in iterative experiments to achieve a balance between accuracy and efficiency.

8. SVM classification: Gaussian RBF, polynomial, quadratic, and linear kernel SVM classifiers were tuned using a fixed train-test split (70–30%). The best-performing kernel was used for the final evaluation.

9. Data splitting strategy: 70% of the data was utilized for training, and 30% for testing using stratified sampling. Additionally, a fixed train-test split was used to avoid instability and overfitting.

10. Evaluation metrics: Model performance was assessed using accuracy, precision, recall, F1-score, AUC-ROC, and confusion matrices for all multi-class experiments.

By using this structured approach, it is possible to integrate deep-learning-based feature extraction, feature selection, and SVM-based classification to build a well-optimized and computationally efficient classification model.

## Pre-trained CNN models

Due to the deep learning advances of recent years, CNNs have become extremely important in realizing state-of-the-art results for various tasks. In this section, we will describe a few popular CNN architectures-EfficientNet, GoogLeNet, ResNet-18, ResNet-50, and ShuffleNet. EfficientNet (*Giroti et al., 2023*) is an important step in capacity estimation between accuracy and computational efficiency.

The model makes use of compound scaling, which scales depth, width, and resolution uniformly in the network, trading off model performance with resources. This method boosts accuracy with a lower computation cost. Additionally, EfficientNet requires compounded hyperparameter tuning, making it both memory-intensive and resource-demanding.

GoogLeNet (Inception VI) from Google and published in 2014 brought significant architectural enhancements in CNN (*Szegedy et al., 2015*) architecture, focusing on efficiency and speed. GoogLeNet lowers the number of parameters compared to its earlier counterparts, such as AlexNet and OxfordNet with a novel design that minimizes the calculations. Subsequently, traces of development in neural architecture were drawn from the parameter efficiency of GoogLeNet.

**Table 2 Extracted features layer, number of layers and size of features for each pre-trained CNN model.**

| Pre-trained CNN models | Number of layers | Extracted features layer | Size of features |
|---|---|---|---|
| Efficientnetb0 | 290 | efficientnet-b0\|model\|head\|global_average_pooling2d\|GlobAvgPool | 1,280 |
| GoogleNet | 144 | pool5-drop_7 × 7_s1 | 1,024 |
| ResNet50 | 177 | avg_pool | 2,048 |
| ResNet18 | 71 | pool5 | 512 |
| Shufflenet | 172 | node_200 | 544 |

The Microsoft ResNet-18 and ResNet-50 (*Tang & Teoh, 2023*; *Hanne et al., 2022*) give rich attention to deep residual learning. The skip/connections allow the networks to bypass the intermediate layers so that the gradient can flow better during backpropagation and reduce vanishing gradients. This favors better training efficiency and accuracy while consuming a heap of computational and memory resources.

ShuffleNet (*Yumang et al., 2023*) maintains computation and memory efficiency while still guaranteeing accuracy. The structure uses group convolution and channel shuffling as means to deal with the computational burden. Group convolution reduces the operations by dividing input channels into groups, while channel shuffling improves performance by rearranging channels. In this work, we will perform feature extraction using these pre-trained convolutional neural network models, as discussed in Table 2.

For stability and convenience, we train the CNN models for 50 epochs at a constant learning rate of 0.01. Stable and repeatable results are produced by the fixed learning rate, which avoids the instability brought on by dynamic changes.

## Support vector machine

One of the most widely used techniques in machine learning is SVM (*Cortes & Vapnik, 1995*). In a generalized high-dimensional space, SVM finds the optimal hyperplane to divide various classes of data points (*Liu et al., 2015*). The transformation process allows for discerning complex relationships and patterns in data, which makes SVM very effective in many applications (*Cristianini & Shawe-Taylor, 2014*).

### Gaussian SVM: medium, coarse, and fine variants

Here, we consider the Gaussian kernel (radial basis function), which is the most popular due to its versatility in coping with non-linear data. Equation (7) provides the Gaussian kernel's mathematical expression.

$$G(x_i, x_j) = \exp\left(\frac{\|x_i - x_j\|^2}{2\sigma^2}\right) = \exp\left(-r\|x_i - x_j\|^2\right) \tag{7}$$

where $\sigma$ is the width of the kernel and determines the flexibility of the decision boundary. The very small $\sigma$ gives overfitting, where a model memorizes noise in the data, and very large $\sigma$ value underfitting where the model fails to capture data complexity. Hence, choosing a suitable kernel width is important. For different versions of Gaussian SVM, the kernel scaling parameter r corresponds to σ\sigmaσ.

The parameter r is defined for fine, medium, and coarse Gaussian SVMs as follows:

$r_{fG} = \sqrt{p/4}$, for fine Gaussian

$r_{mG} = \sqrt{p}$, for a medium Gaussian

$r_{cG} = 4\sqrt{p}$ , for coarse Gaussian

p represents the number of attributes or the dimension of $x_1$. These various Gaussian kernels demonstrate differing levels of algorithmic classification complexity.

- Fine Gaussian: It can be used in cases where there is a need to classify data with complicated decision surfaces.
- Medium Gaussian: It is well-suited to the classification of moderately complex data and provides a compromise between flexibility and simplicity.
- Coarse Gaussian: Effective for low-complexity data, providing a simpler decision boundary.

The selection of the Gaussian kernel needs to be made depending on the complexity of the data, as well as the application needs.

### Cubic SVM

Cubic SVM is another type of SVM that uses a cubic kernel function. This kernel type is useful when the decision boundary has a polynomial nature. The cubic kernel is represented as Eq. (8).

$$K\left(x_i, x_j\right) = \left(x_i^T x_j + 1\right)^3. \tag{8}$$

Cubic SVM proves helpful in situations where there is a need to capture other higher-level relationships between data points. While it might take a big toll on performance compared to linear kernels, Cubic SVM is useful for stressing the need to tilt the complex patterns without needing the performance overhead of using sophisticated kernels like Gaussian RBF.

### Linear SVM

Linear kernels are the basic types of SVM kernels and are normally used when the data is already in a linearly separable form. The linear kernel calculates the inner product of two vectors, $x_i$, and $x_j$, with an optional constant b, as shown in Eq. (9):

$$K\langle x_i, x_j\rangle = x_i^T \cdot x_j + b. \tag{9}$$

This kernel computes the angles between two data points by taking into account the distance between them. Linear SVM tries to be fast and is usually employed for non-complicated large-scale datasets. It does not excel when the data is not linearly separable.

### Quadratic SVM

Quadratic SVM uses polynomial kernels of degree two, which makes it effective for capturing quadratic features of the data. The quadratic kernel is expressed as shown in Eq. (10):

$$K\left(x_i, x_j\right) = 1 - \frac{\|x_i - x_j\|^2}{\|x_i - x_j\|^2 + b}. \tag{10}$$

Compared to the Gaussian kernel, this kernel is less costly in terms of computation and is, therefore, more ideal for application areas, which may otherwise be prohibitive with Gaussian kernels.

Because quadratic forms have important data characteristics, quadratic SVM is widely used in image perception and speech recognition.

### Dimensionality reduction and its impact on SVM

The effectiveness of SVM, like most machine learning techniques, is directly affected by these two features of the dataset: the number of dimensions (d) of the input variables and the number of samples (n). In comparison to low-dimensional ones, high-dimensional datasets usually increase the algorithm's complexity and may be detrimental to its performance. For this reason, the first step with SVM is usually to apply some sort of dimensionality reduction technique like PCA or feature selection. Especially with many features, overfitting is a concern with SVM, and dimensionality reduction helps counter this as well.

## Feature selection methods

Feature selection is important for every machine learning framework, and it can be tricky to find which input variables can produce good predictions for the problem. Reducing the set of attributes through feature selection improves overfitting and model performance and simplifies result interpretation.

### Analysis of variance

Analysis of variance (ANOVA) (*Singh, Kumar & Sagar, 2021*) is a statistical approach used to work out whether there are significant differences among the means of two or more data sets. When it comes to feature selection, ANOVA effectively identifies not just usable features but highly relevant features that strongly correlate with the target variable.

### Chi-square

The chi-square is a statistical method used by researchers to understand the relationship between two variables (*Pan, Liu & Liu, 2020*). The chi-square is sometimes used in feature selection to determine how well a feature by itself correlates with the target variable. If a feature's chi-square value is high, it indicates that the feature probably offers valuable insight into the target variable.

## Feature reduction methods

Feature reduction techniques play a vital role in reducing the original set of attributes to a reduced set of features that retain relevant information and reduces redundant information. This not only reduces the dimensionality of the data, which allows you to build a sparser logic model, but also increases the accuracy and computer efficiency of your ML models. PCA and SVD are the two prevalent feature reduction techniques that we utilize in this work.

### Principal component analysis

A statistical method called principal component analysis (PCA) (*Zeng & Lou, 2020*) can reduce the dimension of large data sets with numerous variables without sacrificing important information. It is common practice to normalize data when scores on certain variables fall on different scales. The next step is to compute a covariance matrix in PCA to express data regarding the relationships between variables. The principal components—new uncorrelated variables that represent the directions of maximum variance in the data—are identified by calculating the eigenvalues and eigenvectors. Effectively reducing the dimensionality of this dataset is achieved by selecting principal components and projecting the data into these additional projections.

### Unified value decomposition or SVD

By breaking down a matrix into its parts, SVD (*Yan et al., 2021*) is a matrix factorization that reveals the underlying data structure.

## RESULTS

Our GoogLeNet-ResNet18 hybrid method addresses a main gap in the literature through a synergistic combination of multiscale (Inception units) and residual features, rather than addressing them separately. The proposed framework achieved 99.78% accuracy through robust feature extraction and testing on actual data beyond the BRACOL dataset. Moreover, we employed PCA/SVD fusion as feature concatenation, while ANOVA and chi-square are enhanced feature selection methods that improve model performance.

## DATABASE utilized

The RoCoLe coffee leaf image dataset (*Parraga-Alava et al., 2019*), was used in this study to perform binary and multi-class classification tasks in this study. The dataset consists of 1,560 coffee leaf images divided into healthy and unhealthy coffee leaves. The dataset is divided into 791 healthy and 769 unhealthy leaves, forming a binary classification. The unhealthy leaves are then classified into five different classes (Rust Level 1 (RL1), Rust Level 2 (RL2), Rust Seated Level (RL3), Rust Level 4 (RL4), and Red Spider Mite (RSM).

## Binary classification

### First experiment

We examined individual CNN models, which use metrics like accuracy, recall, specificity, precision, F-measure, and training time to evaluate the performance of different CNN models (EfficientNetB0, GoogLeNet, ResNet50, ResNet18, and ShuffleNet). With an emphasis on important metrics like accuracy, training time, AUC-ROC, recall, specificity, precision, and F-measure, Table 3 compares the classification performance of several machine learning models.

In Table 3 it shows that EfficientNetb0 performs in a well-balanced manner, while GoogleNet stands out with the highest accuracy. ShuffleNet does well but has somewhat lower metrics, while ResNet50 and ResNet18 show comparable accuracy. While training times vary, GoogleNet is comparatively quicker. AUC-ROC values highlight the

**Table 3 Comparative classification performance of various pre-trained CNN models on the binary database.**

|                | Accuracy % | Training time | AUC-ROC | Recall | Specificity | Precision | F-measure |
|----------------|------------|---------------|---------|--------|-------------|-----------|-----------|
| Efficientnetb0 | 88.88      | 21 min 34 s   | 88.83   | 0.844  | 0.932       | 0.924     | 0.882     |
| GoogleNet      | 93.16      | 12 min 18 s   | 93.13   | 0.905  | 0.958       | 0.954     | 0.929     |
| Resnet50       | 92.3       | 16 min 18 s   | 92.28   | 0.905  | 0.941       | 0.937     | 0.921     |
| Resnet18       | 92.3       | 12 min 3 s    | 92.26   | 0.887  | 0.958       | 0.953     | 0.919     |
| Shufflenet     | 91.02      | 11 min 53 s   | 90.97   | 0.866  | 0.954       | 0.948     | 0.905     |

significance of considering a variety of metrics for a thorough assessment and indicate good overall model performance.

### Second experiment

We extracted features for all images in the dataset from different layers of CNN models, as explained in Table 2. Then we trained SVM kernels (coarse Gaussian, cubic, fine Gaussian, linear, medium Gaussian, and quadratic) for each pre-trained model and studied the performance of this classifier. A thorough examination of the classification performance trained on the CNN models-based feature extraction is shown in Table 4. As inputs to the SVM, this classification model uses a variety of kernel functions for representation learning. With a remarkable 99.8% AUC and 98.07% accuracy using a variety of SVM kernels (cubic, linear, and quadratic), GoogleNet notably outperforms other techniques. Following closely is ResNet18, attaining a 97.64% accuracy with a linear SVM kernel and a commendable training time. Efficientnet, ShuffleNet and ResNet50 are promising models with good performance.

### Third experiment

In this phase of experimentation, the features that were gotten from pairs of CNNs models were combined to have new feature sets. Next, we trained SVMs with different kernels applied to every combined model shown in Table 5. The highest accuracy was achieved by the combination of GoogleNet with ResNet18, which reached 99.14% for both kernels (linear and quadratic) among all combinations studied. Using a linear SVM to detect coffee leaf disease in 3.804 s, the proposed model demonstrated notable improvements, particularly in terms of training time. Furthermore, the model showed remarkable accuracy with a relatively small number of features by exhibiting the following feature dimensions for various model combinations: EfficientNetb + GoogleNet (2,304), EfficientNetb + ResNet18 (1,792), EfficientNetb + ResNet50 (3,328), EfficientNetb + ShuffleNet (1,824), GoogleNet + ResNet18 (1,536), GoogleNet + ResNet50 (3,072), GoogleNet + ShuffleNet (1,568), ResNet18 + ResNet50 (2,560), ResNet18 + ShuffleNet (1,056), and ResNet50 + ShuffleNet (2,592) features. This illustrates the effectiveness of the proposed model in classifying images of coffee leaves with smaller feature sizes.

### Fourth experiment

We analyze our proposed model (GoogleNet + ResNet18) with feature reduction techniques, PCA, and SVD, in combination with various SVM kernels, as shown in

**Table 4 Results for pre-trained CNN models with different SVM Kernels for binary classification.**

| | SVM | Accuracy | Training time | AUC | Recall | Specificity | Precision | F-measure |
|---|---|---|---|---|---|---|---|---|
| EfficientNetb0 | Linear | 95.07% | 7.9281 | 0.9911 | 0.922 | 0.979 | 0.977 | 0.949 |
| | Quadratic | 94.22% | 6.4889 | 0.9897 | 0.913 | 0.97 | 0.968 | 0.94 |
| | Cubic | 94.00% | 10.777 | 0.9896 | 0.909 | 0.97 | 0.968 | 0.938 |
| | Fine Gaussian | 52.89% | 10.566 | 0.8608 | 0.048 | 0.996 | 0.917 | 0.091 |
| | Medium Gaussian | 95.50% | 10.395 | 0.9905 | 0.935 | 0.975 | 0.973 | 0.954 |
| | Coarse Gaussian | 94.22% | 10.677 | 0.9893 | 0.887 | 0.996 | 0.995 | 0.938 |
| GoogleNet | Linear | 98.07% | 3.6647 | 0.9979 | 0.965 | 0.996 | 0.996 | 0.98 |
| | Quadratic | 98.07% | 5.9587 | 0.9982 | 0.974 | 0.987 | 0.987 | 0.98 |
| | Cubic | 98.07% | 5.6103 | 0.998 | 0.97 | 0.992 | 0.991 | 0.98 |
| | Fine Gaussian | 64.45% | 5.387 | 0.9622 | 0.283 | 0.996 | 0.985 | 0.44 |
| | Medium Gaussian | 97.00% | 5.2238 | 0.9957 | 0.97 | 0.97 | 0.97 | 0.97 |
| | Coarse Gaussian | 98.07% | 5.0248 | 0.9979 | 0.961 | 1 | 1 | 0.98 |
| ResNet50 | Linear | 97.43% | 4.9695 | 0.9939 | 0.961 | 0.987 | 0.987 | 0.974 |
| | Quadratic | 97.22% | 9.8943 | 0.9935 | 0.957 | 0.987 | 0.987 | 0.972 |
| | Cubic | 97.22% | 9.4065 | 0.9937 | 0.961 | 0.983 | 0.982 | 0.971 |
| | Fine Gaussian | 53.75% | 18.223 | 0.941 | 0.061 | 1 | 1 | 0.115 |
| | Medium Gaussian | 97.43% | 8.2295 | 0.9918 | 0.965 | 0.983 | 0.982 | 0.973 |
| | Coarse Gaussian | 97.00% | 8.2277 | 0.9928 | 0.952 | 0.987 | 0.986 | 0.969 |
| ResNet18 | Linear | 97.64% | 2.8068 | 0.9945 | 0.965 | 0.987 | 0.987 | 0.976 |
| | Quadratic | 97.43% | 3.5247 | 0.994 | 0.965 | 0.983 | 0.982 | 0.973 |
| | Cubic | 97.22% | 3.1817 | 0.9929 | 0.961 | 0.983 | 0.982 | 0.971 |
| | Fine Gaussian | 54.18% | 2.9716 | 0.968 | 0.074 | 0.996 | 0.944 | 0.137 |
| | Medium Gaussian | 97.43% | 2.8248 | 0.9941 | 0.965 | 0.983 | 0.982 | 0.973 |
| | Coarse Gaussian | 97.22% | 4.1663 | 0.9958 | 0.961 | 0.983 | 0.982 | 0.971 |
| ShuffleNet | Linear | 96.15% | 3.0525 | 0.9935 | 0.939 | 0.983 | 0.982 | 0.96 |
| | Quadratic | 96.36% | 1.4526 | 0.9931 | 0.943 | 0.983 | 0.982 | 0.962 |
| | Cubic | 95.93% | 3.1126 | 0.9935 | 0.939 | 0.979 | 0.977 | 0.958 |
| | Fine Gaussian | 53.10% | 2.9357 | 0.9272 | 0.048 | 1 | 1 | 0.092 |
| | Medium Gaussian | 96.36% | 2.7674 | 0.9945 | 0.948 | 0.979 | 0.978 | 0.963 |
| | Coarse Gaussian | 95.50% | 2.6001 | 0.9943 | 0.926 | 0.983 | 0.982 | 0.953 |

**Table 5 Results obtained using combining each two CNN models with SVM kernels for binary classification.**

| | SVM | ACC | Training time | AUC | Recall | Specificity | Precision | F-measure |
|---|---|---|---|---|---|---|---|---|
| EfficientNetb +GoogleNet | Linear | 97.64% | 16.883 | 0.9976 | 0.957 | 0.996 | 0.995 | 0.976 |
| | Quadratic | 97.22% | 22.706 | 0.9964 | 0.952 | 0.992 | 0.991 | 0.971 |
| | Cubic | 97.22% | 20.624 | 0.9962 | 0.952 | 0.992 | 0.991 | 0.971 |
| | Fine Gaussian | 54.39% | 43.765 | 0.9474 | 0.074 | 1 | 1 | 0.138 |
| | Medium Gaussian | 97.86% | 20.529 | 0.995 | 0.965 | 0.992 | 0.991 | 0.978 |
| | Coarse Gaussian | 97.43% | 21.136 | 0.999 | 0.948 | 1 | 1 | 0.973 |

|  | SVM | ACC | Training time | AUC | Recall | Specificity | Precision | F-measure |
|---|---|---|---|---|---|---|---|---|
| EfficientNetb +ResNet18 | Linear | 98.50% | 7.4026 | 0.996 | 0.978 | 0.992 | 0.991 | 0.984 |
|  | Quadratic | 98.07% | 12.128 | 0.9958 | 0.974 | 0.987 | 0.987 | 0.98 |
|  | Cubic | 97.86% | 10.234 | 0.9951 | 0.974 | 0.983 | 0.982 | 0.978 |
|  | Fine Gaussian | 53.53% | 12.839 | 0.9453 | 0.061 | 0.996 | 0.933 | 0.115 |
|  | Medium Gaussian | 97.86% | 9.7431 | 0.9943 | 0.97 | 0.987 | 0.987 | 0.978 |
|  | Coarse Gaussian | 97.86% | 12.693 | 0.9962 | 0.965 | 0.992 | 0.991 | 0.978 |
| EfficientNetb +ResNet50 | Linear | 97.43% | 14.563 | 0.9972 | 0.961 | 0.987 | 0.987 | 0.974 |
|  | Quadratic | 98.29% | 18.382 | 0.9976 | 0.974 | 0.992 | 0.991 | 0.982 |
|  | Cubic | 98.07% | 16.172 | 0.9977 | 0.965 | 0.996 | 0.996 | 0.98 |
|  | Fine Gaussian | 53.75% | 43.115 | 0.8904 | 0.061 | 1 | 1 | 0.115 |
|  | Medium Gaussian | 97.64% | 22.693 | 0.9955 | 0.965 | 0.987 | 0.987 | 0.976 |
|  | Coarse Gaussian | 96.57% | 23.699 | 0.9954 | 0.935 | 0.996 | 0.995 | 0.964 |
| EfficientNetb +ShuffleNet | Linear | 97.00% | 7.938 | 0.9979 | 0.961 | 0.979 | 0.978 | 0.969 |
|  | Quadratic | 97.22% | 14.691 | 0.9978 | 0.97 | 0.975 | 0.974 | 0.972 |
|  | Cubic | 96.57% | 12.582 | 0.9969 | 0.961 | 0.97 | 0.969 | 0.965 |
|  | Fine Gaussian | 54.39% | 15.112 | 0.9285 | 0.078 | 0.996 | 0.947 | 0.144 |
|  | Medium Gaussian | 96.36% | 11.388 | 0.9971 | 0.957 | 0.97 | 0.969 | 0.963 |
|  | Coarse Gaussian | 95.07% | 11.969 | 0.9964 | 0.922 | 0.979 | 0.977 | 0.949 |
| GoogleNet +ResNet18 | Linear | 99.14% | 3.8048 | 0.9992 | 0.987 | 0.996 | 0.996 | 0.991 |
|  | Quadratic | 99.14% | 6.343 | 0.9991 | 0.987 | 0.996 | 0.996 | 0.991 |
|  | Cubic | 98.50% | 7.892 | 0.9991 | 0.978 | 0.992 | 0.991 | 0.984 |
|  | Fine Gaussian | 54.60% | 10.531 | 0.9761 | 0.078 | 1 | 1 | 0.145 |
|  | Medium Gaussian | 98.29% | 6.5486 | 0.9974 | 0.978 | 0.987 | 0.987 | 0.982 |
|  | Coarse Gaussian | 98.07% | 6.5653 | 0.9989 | 0.97 | 0.992 | 0.991 | 0.98 |
| GoogleNet +ResNet50 | Linear | 98.93% | 11.696 | 0.9994 | 0.983 | 0.996 | 0.996 | 0.989 |
|  | Quadratic | 99.14% | 15.925 | 0.9992 | 0.983 | 1 | 1 | 0.991 |
|  | Cubic | 99.14% | 13.316 | 0.9992 | 0.983 | 1 | 1 | 0.991 |
|  | Fine Gaussian | 54.82% | 46.062 | 0.9766 | 0.083 | 1 | 1 | 0.153 |
|  | Medium Gaussian | 98.93% | 17.348 | 0.9979 | 0.983 | 0.996 | 0.996 | 0.989 |
|  | Coarse Gaussian | 98.93% | 18.231 | 0.9992 | 0.983 | 0.996 | 0.996 | 0.989 |
| GoogleNet +ShuffleNet | Linear | 97.43% | 5.1053 | 0.9955 | 0.961 | 0.987 | 0.987 | 0.974 |
|  | Quadratic | 97.86% | 6.046 | 0.9963 | 0.97 | 0.987 | 0.987 | 0.978 |
|  | Cubic | 97.64% | 4.8812 | 0.9966 | 0.965 | 0.987 | 0.987 | 0.976 |
|  | Fine Gaussian | 54.39% | 27.67 | 0.9677 | 0.074 | 1 | 1 | 0.138 |
|  | Medium Gaussian | 97.43% | 6.7525 | 0.992 | 0.965 | 0.983 | 0.982 | 0.973 |
|  | Coarse Gaussian | 97.22% | 6.1196 | 0.9944 | 0.952 | 0.992 | 0.991 | 0.971 |
| ResNet18 +ResNet50 | Linear | 98.07% | 9.1408 | 0.9973 | 0.97 | 0.992 | 0.991 | 0.98 |
|  | Quadratic | 98.50% | 13.936 | 0.9974 | 0.974 | 0.996 | 0.996 | 0.985 |
|  | Cubic | 98.50% | 11.816 | 0.9975 | 0.978 | 0.992 | 0.991 | 0.984 |
|  | Fine Gaussian | 53.32% | 19.099 | 0.9756 | 0.052 | 1 | 1 | 0.099 |
|  | Medium Gaussian | 98.07% | 13.571 | 0.9979 | 0.97 | 0.992 | 0.991 | 0.98 |
|  | Coarse Gaussian | 98.07% | 14.417 | 0.9973 | 0.97 | 0.992 | 0.991 | 0.98 |

(Continued)

| | SVM | ACC | Training time | AUC | Recall | Specificity | Precision | F-measure |
|---|---|---|---|---|---|---|---|---|
| ResNet18 +ShuffleNet | Linear | 98.72% | 3.6578 | 0.9983 | 0.987 | 0.987 | 0.987 | 0.987 |
| | Quadratic | 98.50% | 4.4816 | 0.9981 | 0.987 | 0.983 | 0.983 | 0.985 |
| | Cubic | 98.72% | 6.5822 | 0.998 | 0.991 | 0.983 | 0.983 | 0.987 |
| | Fine Gaussian | 53.32% | 25.799 | 0.9739 | 0.061 | 0.992 | 0.875 | 0.114 |
| | Medium Gaussian | 98.72% | 4.7959 | 0.9972 | 0.987 | 0.987 | 0.987 | 0.987 |
| | Coarse Gaussian | 98.72% | 3.7785 | 0.998 | 0.983 | 0.992 | 0.991 | 0.987 |
| ResNet50 +ShuffleNet | Linear | 98.93% | 10.018 | 0.9991 | 0.987 | 0.992 | 0.991 | 0.989 |
| | Quadratic | 98.72% | 15.859 | 0.9988 | 0.983 | 0.992 | 0.991 | 0.987 |
| | Cubic | 98.93% | 13.221 | | 0.987 | 0.992 | 0.991 | 0.989 |
| | Fine Gaussian | 54.39% | 18.091 | 0.9668 | 0.078 | 0.996 | 0.947 | 0.144 |
| | Medium Gaussian | 98.93% | 14.259 | 0.999 | 0.987 | 0.992 | 0.991 | 0.989 |
| | Coarse Gaussian | 98.50% | 14.89 | 0.9986 | 0.978 | 0.992 | 0.991 | 0.984 |

**Table 6 Results of proposed model with PCA and SVD based feature reduction methods for binary classification.**

| | SVM | ACC % | ER | RE | SP | P | F-measure | Training time | Testing time |
|---|---|---|---|---|---|---|---|---|---|
| PCA based proposed | Linear | 95.075 | 4.925 | 0.93 | 0.97 | 0.968 | 0.949 | 0.083 | 0.017 |
| | Quadratic | 96.146 | 3.854 | 0.943 | 0.979 | 0.977 | 0.96 | 0.1 | 0.017 |
| | Cubic | 94.861 | 5.139 | 0.917 | 0.979 | 0.977 | 0.946 | 0.089 | 0.016 |
| | Fine Gaussian | 53.747 | 46.253 | 0.061 | 1 | 1 | 0.115 | 4.718 | 0.029 |
| | Medium Gaussian | 87.794 | 12.206 | 0.926 | 0.831 | 0.842 | 0.882 | 0.104 | 0.037 |
| | Coarse Gaussian | 50.749 | 49.251 | 0 | 1 | NaN | NaN | 0.09 | 0.02 |
| SVD based proposed | Linear | 98.501 | 1.499 | 0.974 | 0.996 | 0.996 | 0.985 | 0.258 | 0.019 |
| | Quadratic | 98.929 | 1.071 | 0.983 | 0.996 | 0.996 | 0.989 | 0.23 | 0.025 |
| | Cubic | 98.501 | 1.499 | 0.974 | 0.996 | 0.996 | 0.985 | 0.308 | 0.025 |
| | Fine Gaussian | 54.176 | 45.824 | 0.07 | 1 | 1 | 0.13 | 0.837 | 0.21 |
| | Medium Gaussian | 93.79 | 6.21 | 0.974 | 0.903 | 0.907 | 0.939 | 0.642 | 0.058 |
| | Coarse Gaussian | 97.645 | 2.355 | 0.957 | 0.996 | 0.995 | 0.976 | 1.031 | 0.231 |

Table 6. The PCA-based model achieved 96.146% accuracy with the quadratic SVM, demonstrating strong performance across several metrics, including recall, specificity, precision, and F-measure, especially with linear SVM. The SVD model outperforms the quadratic model, yielding 98.929% accuracy and a very well-balanced trade-off between precision and recall using linear SVM while maintaining excellent performance on multiple metrics. Furthermore, the dimensionality of the combined datasets from GoogleNet and ResNet18 has been efficiently reduced through the implementation of PCA and SVD feature reduction techniques. Specifically, PCA reduced the feature set from 1,536 to 632 features, while SVD reduced it to 1,185 features. Crucially, both methods achieved this reduction while maintaining 99.99% of the information in the original dataset. This high degree of information preservation shows how well PCA and SVD capture the key characteristics that greatly influence the variance of the dataset. The slight

**Table 7 Results of proposed model with chi-square and anova based feature selection methods for binary classification.**

|  | SVM | ACC % | ER | RE | SP | P | FM | Training time | Testing time |
|---|---|---|---|---|---|---|---|---|---|
| Anova based proposed | Linear | 99.572 | 0.428 | 0.991 | 1 | 1 | 0.996 | 0.069 | 0.028 |
|  | Quadratic | 99.572 | 0.428 | 0.991 | 1 | 1 | 0.996 | 0.062 | 0.031 |
|  | Cubic | 99.572 | 0.428 | 0.991 | 1 | 1 | 0.996 | 0.096 | 0.038 |
|  | Fine Gaussian | 55.889 | 44.111 | 0.104 | 1 | 1 | 0.189 | 5.277 | 0.044 |
|  | Medium Gaussian | 99.786 | 0.214 | 0.996 | 1 | 1 | 0.998 | 0.073 | 0.029 |
|  | Coarse Gaussian | 98.501 | 1.499 | 0.97 | 1 | 1 | 0.985 | 0.308 | 0.127 |
| Chi-Square based proposed | Linear | 98.073 | 1.927 | 0.965 | 0.996 | 0.996 | 0.98 | 0.084 | 0.032 |
|  | Quadratic | 98.501 | 1.499 | 0.974 | 0.996 | 0.996 | 0.985 | 0.06 | 0.034 |
|  | Cubic | 98.073 | 1.927 | 0.965 | 0.996 | 0.996 | 0.98 | 0.119 | 0.034 |
|  | Fine Gaussian | 60.6 | 39.4 | 0.2 | 1 | 1 | 0.333 | 0.199 | 0.056 |
|  | Medium Gaussian | 98.073 | 1.927 | 0.965 | 0.996 | 0.996 | 0.98 | 0.075 | 0.027 |
|  | Coarse Gaussian | 97.645 | 2.355 | 0.957 | 0.996 | 0.995 | 0.976 | 0.309 | 0.082 |

reduction in accuracy (0.5–1%) following PCA/SVD is because of the removal of low-variance but discriminating features. The trade-off between dimensionality reduction and feature preservation was made carefully, with 99% variance selected for preserving important patterns at the cost of minimal computational overhead.

### Fifth experiment

We used ANOVA and chi-square techniques to refine the features of the selected model (GoogleNet + ResNet18), as shown in Table 7. The proposed model demonstrated the superior accuracy of ANOVA-based feature selection with an accuracy of 99.78%, an error rate of 0.214, and a recall of 0.996 when paired with a medium Gaussian SVM kernel. Similarly, when combined with a medium Gaussian SVM kernel, the chi-square-based feature selection method produced excellent results, with an accuracy of 98.073%, an error rate of 1.927%, and a recall of 0.965. The matching times for testing and training were 0.027 and 0.075 s, respectively.

From these findings, it emerges clearly that both ANOVA and chi-square play critical roles in improving the model's accuracy and performance. However, the approach based on ANOVA performed better than chi-square in accuracy and recall. Knowing that the number of features is reduced to 500 for both ANOVA and chi-square feature selection methods. Figure 2 shows feature weights using the Anova and chi-square methods for feature selection in a proposed model for binary classification.

In the initial experiment, we demonstrated that the proposed model has significantly improved accuracy, from 93.16% to 98.07% using feature reduction (PCA and SVD) which maintained high levels of accuracy. The use of ANOVA feature selection recorded an impressive accuracy rate of 99.78%. In addition, the proposed model led to a great decrease in feature dimensions from 1,536 to 500 which proved its effectiveness for a smaller set of discriminative features. This in-depth analysis thus confirms that the proposed model is effective in promoting the classification of coffee leaf images, constituting a valuable contribution to image recognition and disease diagnosis.

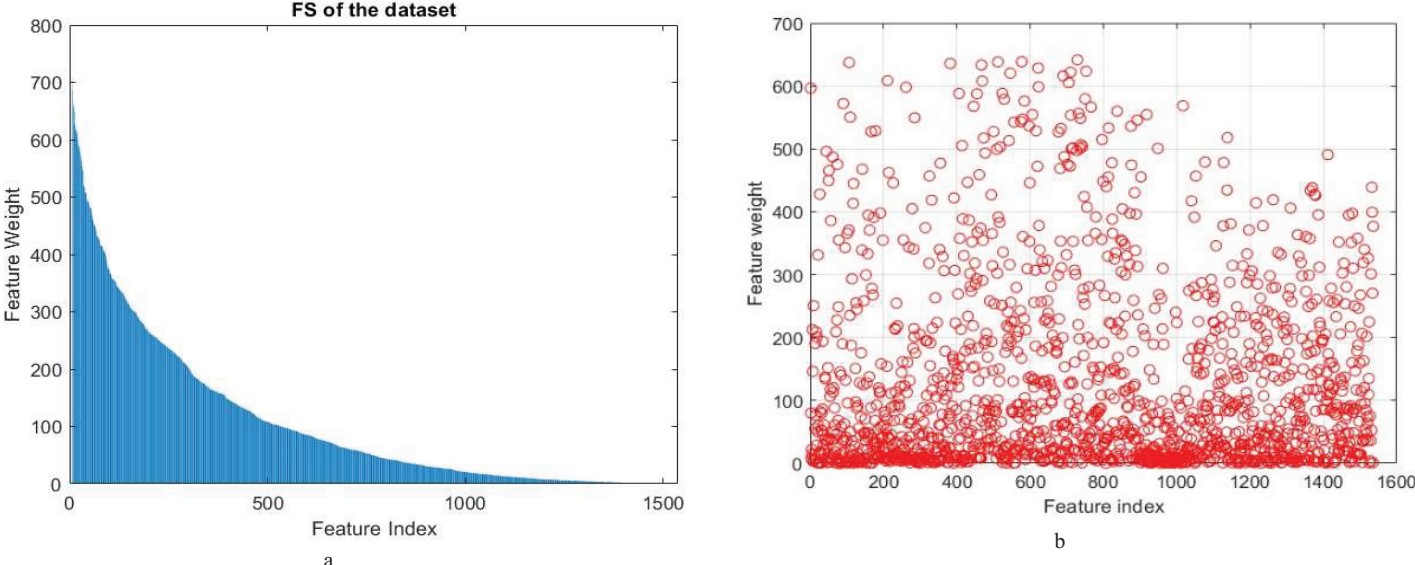

**Figure 2 (A) Feature weight of (A) ANOVA (B) chi-square feature selection methods of proposed model for binary classification.**

## Multi-level classification

Model performances in multi-level classification will be assessed based on accuracy, training time, AUC-ROC, class-specific accuracies for each class like "healthy", "RSM", "RL1", "RL2", "RL3", and "RL4". The separate analysis of accuracy for each class allows us to know how well a model performs in certain classes and understand its possible advantages or disadvantages for specific ones.

### First experiment

Further analysis was conducted on the multi-class classification results, which are given in Table 8 demonstrates different performance experiments for various pre-trained CNN models that have been tested. ResNet18 is the best model as it records the highest accuracy of 72.86% and a stronger AUC-ROC value of 0.9245%. GoogleNet and ShuffleNet present competing accuracies of 70.51% and 70.08%, GoogleNet additionally shows significant time efficiency while training (15 min 26 s). ResNet50 has an accuracy that is not far with 69.44% while its training time took a short period of minutes and seconds calculated to be 14:51 s. Some models share mutual performance for some classes; hence the ResNet18 and GoogleNet are more robust than others. Depending on particular needs, selecting the best model involves some trade-offs in various domains, such as accuracy *vs.* training time efficiency *vs.* AUC-ROC performance for enhanced multi-level classification.

### Second experiment

A detailed analysis of multi-class classification based on SVM classifiers with different kernels combined with different CNN models was conducted in our work, and several performance features in Table 9. Linear SVM has high accuracy consistently across EfficientNetb0, ResNet50, resNet18, GoogleNet and ShuffleNet models with the highest

**Table 8 Comparative classification performance of various pretrained CNN models on multi-class database.**

|  | Accuracy | Training time | AUC-ROC | Healthy | RSM | RL1 | RL2 | RL3 | RL4 |
|---|---|---|---|---|---|---|---|---|---|
| Efficientnetb0 | 66.23% | 74 min 38 s | 0.842 | 0.932 | 0.34 | 0.447 | 0.44 | 0.158 | 0.111 |
| Resnet50 | 69.44% | 14 min 51 s | 0.8788 | 0.928 | 0.4 | 0.427 | 0.8 | 0.053 | 0 |
| Resnet18 | 72.86% | 32 min 53 s | 0.9245 | 0.962 | 0.38 | 0.544 | 0.62 | 0.211 | 0.333 |
| GoogleNet | 70.51% | 15 min 26 s | 0.8932 | 0.966 | 0.5 | 0.456 | 0.56 | 0.053 | 0 |
| Shufflenet | 70.08% | 15 min 25 s | 0.8839 | 0.958 | 0.36 | 0.485 | 0.56 | 0.158 | 0.222 |

**Table 9 Results for pre-trained CNN models with different SVM kernels for multi-class database.**

|  | SVM | ACC | Training time | AUC | Healthy | RSM | RL1 | RL2 | RL3 | RL4 |
|---|---|---|---|---|---|---|---|---|---|---|
| EfficientNetb0 | Linear | 82.01% | 18.184 | 0.9809 | 0.975 | 0.62 | 0.796 | 0.72 | 0.105 | 0.111 |
|  | Quadratic | 83.51% | 23.984 | 0.9812 | 0.962 | 0.58 | 0.796 | 0.72 | 0.632 | 0.333 |
|  | Cubic | 83.73% | 21.748 | 0.9805 | 0.958 | 0.58 | 0.825 | 0.7 | 0.632 | 0.333 |
|  | Fine Gaussian | 51.18% | 22.02 | 0.575 | 1 | 0 | 0.01 | 0.02 | 0 | 0 |
|  | Medium Gaussian | 80.51% | 24.695 | 0.9812 | 0.97 | 0.56 | 0.816 | 0.62 | 0.053 | 0.222 |
|  | Coarse Gaussian | 69.16% | 24.778 | 0.972 | 0.996 | 0 | 0.748 | 0.2 | 0 | 0 |
| ResNet50 | Linear | 85.44% | 9.0085 | 0.9907 | 0.966 | 0.7 | 0.845 | 0.78 | 0.368 | 0.222 |
|  | Quadratic | 84.15% | 12.638 | 0.9899 | 0.966 | 0.7 | 0.796 | 0.7 | 0.474 | 0.333 |
|  | Cubic | 83.73% | 12.01 | 0.9886 | 0.97 | 0.68 | 0.786 | 0.7 | 0.474 | 0.222 |
|  | Fine Gaussian | 52.25% | 19.542 | 0.7696 | 1 | 0 | 0.039 | 0.06 | 0 | 0 |
|  | Medium Gaussian | 81.37% | 10.698 | 0.9892 | 0.97 | 0.68 | 0.728 | 0.72 | 0.211 | 0.111 |
|  | Coarse Gaussian | 75.59% | 10.668 | 0.991 | 0.962 | 0.5 | 0.893 | 0.16 | 0 | 0 |
| ResNet18 | Linear | 88.65% | 45.791 | 0.9901 | 0.992 | 0.82 | 0.835 | 0.82 | 0.474 | 0.222 |
|  | Quadratic | 87.58% | 42.755 | 0.9915 | 0.992 | 0.72 | 0.825 | 0.82 | 0.474 | 0.333 |
|  | Cubic | 87.58% | 35.979 | 0.9898 | 0.992 | 0.72 | 0.845 | 0.8 | 0.474 | 0.222 |
|  | Fine Gaussian | 50.70% | 57.14 | 0.6321 | 1 | 0 | 0 | 0 | 0 | 0 |
|  | Medium Gaussian | 85.44% | 56.38 | 0.9936 | 0.97 | 0.82 | 0.816 | 0.72 | 0.316 | 0.222 |
|  | Coarse Gaussian | 78.60% | 55.08 | 0.9921 | 0.97 | 0.5 | 0.893 | 0.4 | 0 | 0 |
| GoogleNet | Linear | 88.22% | 7.3245 | 0.9935 | 0.962 | 0.78 | 0.883 | 0.8 | 0.474 | 0.556 |
|  | Quadratic | 88.01% | 5.7846 | 0.9937 | 0.966 | 0.76 | 0.883 | 0.78 | 0.474 | 0.556 |
|  | Cubic | 87.79% | 5.0997 | 0.9944 | 0.966 | 0.76 | 0.883 | 0.76 | 0.474 | 0.556 |
|  | Fine Gaussian | 51.39% | 10.017 | 0.9157 | 1 | 0 | 0.019 | 0.02 | 0 | 0 |
|  | Medium Gaussian | 88.22% | 9.6486 | 0.9942 | 0.97 | 0.74 | 0.913 | 0.8 | 0.368 | 0.444 |
|  | Coarse Gaussian | 85.87% | 7.3907 | 0.9949 | 0.966 | 0.74 | 0.932 | 0.72 | 0 | 0 |
| ShuffleNet | Linear | 84.37% | 7.2032 | 0.986 | 0.954 | 0.74 | 0.825 | 0.72 | 0.421 | 0.222 |
|  | Quadratic | 84.15% | 9.4222 | 0.983 | 0.945 | 0.76 | 0.786 | 0.78 | 0.316 | 0.556 |
|  | Cubic | 83.51% | 9.0008 | 0.9833 | 0.941 | 0.72 | 0.786 | 0.78 | 0.316 | 0.556 |
|  | Fine Gaussian | 51.18% | 8.7156 | 0.8754 | 1 | 0 | 0.01 | 0.02 | 0 | 0 |
|  | Medium Gaussian | 82.87% | 8.4397 | 0.9855 | 0.962 | 0.72 | 0.825 | 0.68 | 0.158 | 0.111 |
|  | Coarse Gaussian | 77.73% | 7.4357 | 0.9862 | 0.962 | 0.34 | 0.922 | 0.46 | 0 | 0 |

**Table 10 Results obtained using combining each two CNN models with SVM kernels for multi-class classification.**

| | SVM | ACC | Training time | AUC | Healthy | RSM | RL1 | RL2 | RL3 | RL4 |
|---|---|---|---|---|---|---|---|---|---|---|
| EfficientNetb +GoogleNet | Linear | 88.65% | 19.657 | 0.9965 | 0.979 | 0.82 | 0.864 | 0.84 | 0.368 | 0.333 |
| | Quadratic | 89.51% | 27.087 | 0.996 | 0.979 | 0.82 | 0.845 | 0.86 | 0.579 | 0.444 |
| | Cubic | 89.29% | 25.637 | 0.9957 | 0.979 | 0.82 | 0.845 | 0.84 | 0.579 | 0.444 |
| | Fine Gaussian | 50.96% | 36.916 | 0.5977 | 1 | 0 | 0.01 | 0 | 0 | 0 |
| | Medium Gaussian | 86.51% | 34.177 | 0.9951 | 0.983 | 0.78 | 0.835 | 0.76 | 0.316 | 0.222 |
| | Coarse Gaussian | 80.73% | 33.751 | 0.9957 | 0.975 | 0.52 | 0.951 | 0.44 | 0 | 0 |
| EfficientNetb +ResNet18 | Linear | 85.65% | 15.411 | 0.9973 | 0.983 | 0.7 | 0.903 | 0.76 | 0.053 | 0 |
| | Quadratic | 87.79% | 21.877 | 0.9976 | 0.987 | 0.74 | 0.903 | 0.78 | 0.263 | 0.222 |
| | Cubic | 88.01% | 20.222 | 0.9977 | 0.987 | 0.76 | 0.913 | 0.78 | 0.263 | 0.111 |
| | Fine Gaussian | 50.96% | 31.322 | 0.761 | 1.004 | 0 | 0 | 0 | 0 | 0 |
| | Medium Gaussian | 83.94% | 24.102 | 0.9981 | 0.983 | 0.58 | 0.932 | 0.66 | 0.053 | 0 |
| | Coarse Gaussian | 72.59% | 23.148 | 0.9966 | 0.987 | 0.02 | 0.981 | 0.06 | 0 | 0 |
| EfficientNetb +ResNet50 | Linear | 86.94% | 29.785 | 0.9954 | 0.962 | 0.74 | 0.903 | 0.76 | 0.316 | 0.444 |
| | Quadratic | 87.15% | 38.609 | 0.993 | 0.962 | 0.72 | 0.903 | 0.76 | 0.368 | 0.556 |
| | Cubic | 87.15% | 36.478 | 0.9906 | 0.966 | 0.72 | 0.903 | 0.76 | 0.316 | 0.556 |
| | Fine Gaussian | 50.96% | 65.714 | 0.5288 | 1 | 0 | 0 | 0 | 0 | 0 |
| | Medium Gaussian | 85.22% | 58.503 | 0.991 | 0.966 | 0.68 | 0.922 | 0.7 | 0.105 | 0.333 |
| | Coarse Gaussian | 77.94% | 58.372 | 0.9921 | 0.979 | 0.32 | 0.951 | 0.36 | 0 | 0 |
| EfficientNetb +ShuffleNet | Linear | 88.01% | 16.824 | 0.9981 | 0.987 | 0.84 | 0.922 | 0.76 | 0.105 | 0 |
| | Quadratic | 89.08% | 24.834 | 0.9981 | 0.983 | 0.86 | 0.942 | 0.7 | 0.368 | 0.111 |
| | Cubic | 88.01% | 22.848 | 0.9982 | 0.983 | 0.84 | 0.942 | 0.62 | 0.368 | 0.111 |
| | Fine Gaussian | 50.75% | 31.05 | 0.7173 | 1 | 0 | 0 | 0 | 0 | 0 |
| | Medium Gaussian | 84.58% | 29.22 | 0.9982 | 0.975 | 0.72 | 0.932 | 0.62 | 0.053 | 0 |
| | Coarse Gaussian | 72.59% | 26.989 | 0.9966 | 0.987 | 0 | 0.971 | 0.1 | 0 | 0 |
| GoogleNet +ResNet18 | Linear | 91.86% | 27.24 | 0.9997 | 0.987 | 0.86 | 0.971 | 0.8 | 0.632 | 0 |
| | Quadratic | 92.51% | 52.836 | 0.9998 | 0.987 | 0.86 | 0.971 | 0.8 | 0.684 | 0.222 |
| | Cubic | 92.08% | 51.058 | 0.9998 | 0.987 | 0.84 | 0.971 | 0.78 | 0.684 | 0.222 |
| | Fine Gaussian | 50.75% | 49.037 | 0.7054 | 1 | 0 | 0 | 0 | 0 | 0 |
| | Medium Gaussian | 89.08% | 46.396 | 0.9997 | 0.987 | 0.78 | 0.981 | 0.72 | 0.316 | 0 |
| | Coarse Gaussian | 80.51% | 47.229 | 0.9993 | 0.987 | 0.4 | 0.99 | 0.4 | 0 | 0 |
| GoogleNet +ResNet50 | Linear | 89.29% | 54.872 | 0.9995 | 0.966 | 0.82 | 0.922 | 0.82 | 0.474 | 0.222 |
| | Quadratic | 89.94% | 52.475 | 0.9959 | 0.966 | 0.82 | 0.922 | 0.8 | 0.632 | 0.333 |
| | Cubic | 89.51% | 49.499 | 0.9961 | 0.97 | 0.8 | 0.913 | 0.8 | 0.632 | 0.222 |
| | Fine Gaussian | 51.18% | 69.921 | 0.7733 | 1 | 0 | 0 | 0.04 | 0 | 0 |
| | Medium Gaussian | 87.58% | 68.342 | 0.9946 | 0.983 | 0.8 | 0.922 | 0.68 | 0.263 | 0.222 |
| | Coarse Gaussian | 82.01% | 66.482 | 0.9956 | 0.975 | 0.54 | 0.951 | 0.54 | 0 | 0 |
| GoogleNet +ShuffleNet | Linear | 87.37% | 13.458 | 0.9977 | 0.97 | 0.68 | 0.951 | 0.84 | 0.211 | 0 |
| | Quadratic | 87.79% | 21.62 | 0.9978 | 0.975 | 0.66 | 0.951 | 0.78 | 0.474 | 0 |
| | Cubic | 87.58% | 20.314 | 0.9977 | 0.975 | 0.68 | 0.942 | 0.76 | 0.474 | 0 |
| | Fine Gaussian | 50.96% | 21.689 | 0.8616 | 1 | 0 | 0 | 0 | 0 | 0 |
| | Medium Gaussian | 85.44% | 19.784 | 0.9949 | 0.979 | 0.66 | 0.942 | 0.68 | 0.105 | 0.111 |
| | Coarse Gaussian | 75.16% | 18.052 | 0.9965 | 0.962 | 0.4 | 0.981 | 0.04 | 0 | 0 |

|  | SVM | ACC | Training time | AUC | Healthy | RSM | RL1 | RL2 | RL3 | RL4 |
|---|---|---|---|---|---|---|---|---|---|---|
| ResNet18 +ResNet50 | Linear | 89.72% | 13.731 | 0.9993 | 0.983 | 0.84 | 0.883 | 0.86 | 0.421 | 0.222 |
|  | Quadratic | 90.15% | 25.747 | 0.9995 | 0.987 | 0.84 | 0.854 | 0.88 | 0.474 | 0.444 |
|  | Cubic | 89.51% | 24.228 | 0.9995 | 0.992 | 0.82 | 0.845 | 0.86 | 0.474 | 0.333 |
|  | Fine Gaussian | 50.75% | 28.651 | 0.8401 | 1 | 0 | 0 | 0 | 0 | 0 |
|  | Medium Gaussian | 87.37% | 20.198 | 0.9982 | 0.996 | 0.8 | 0.883 | 0.76 | 0.105 | 0.111 |
|  | Coarse Gaussian | 77.30% | 22.54 | 0.9989 | 0.966 | 0.58 | 0.981 | 0.04 | 0 | 0 |
| ResNet18 +ShuffleNet | Linear | 89.51% | 9.84 | 0.9991 | 0.979 | 0.74 | 0.922 | 0.8 | 0.632 | 0.222 |
|  | Quadratic | 89.08% | 15.324 | 0.9993 | 0.979 | 0.68 | 0.922 | 0.8 | 0.632 | 0.333 |
|  | Cubic | 89.72% | 14.582 | 0.9992 | 0.992 | 0.68 | 0.932 | 0.8 | 0.632 | 0.222 |
|  | Fine Gaussian | 51.18% | 18.866 | 0.8571 | 1 | 0 | 0.01 | 0.02 | 0 | 0 |
|  | Medium Gaussian | 87.15% | 13.015 | 0.9988 | 0.983 | 0.66 | 0.903 | 0.76 | 0.474 | 0.111 |
|  | Coarse Gaussian | 84.37% | 11.617 | 0.9987 | 0.983 | 0.58 | 0.942 | 0.7 | 0 | 0 |
| ResNet50 +ShuffleNet | Linear | 90.15% | 28.289 | 0.9945 | 0.987 | 0.74 | 0.913 | 0.8 | 0.737 | 0.222 |
|  | Quadratic | 90.36% | 41.441 | 0.9934 | 0.992 | 0.72 | 0.893 | 0.84 | 0.789 | 0.222 |
|  | Cubic | 89.72% | 39.676 | 0.9927 | 0.992 | 0.72 | 0.903 | 0.78 | 0.737 | 0.222 |
|  | Fine Gaussian | 50.70% | 59.49 | 0.7819 | 1 | 0 | 0 | 0 | 0 | 0 |
|  | Medium Gaussian | 87.80% | 57.792 | 0.9944 | 0.987 | 0.7 | 0.874 | 0.88 | 0.316 | 0.111 |
|  | Coarse Gaussian | 79.90% | 56.069 | 0.9932 | 0.979 | 0.44 | 0.942 | 0.44 | 0 | 0 |

occurring in the case of Reesrnt81 at 88.65%. Quadratic and Cubic SVM kernels also yield competitive performances in multiple models. Notably, the Fine Gaussian SVM kernel exhibits suboptimal accuracy, particularly evident in ResNet18 and GoogleNet models.

### Third experiment

The results are presented in Table 10, describing the operation of combining features from pairs of pre-trained CNN models, followed by SVM training with a variety of kernels, in the exploration of multi-level classification. Especially noteworthy were combinations ResNet50+ShuffleNet and ResNet18+ResNet50: both achieved over 90% accuracy, deriving features from 2,592 and 2,560 units, respectively. GoogleNet+ResNet18 stands out, achieving a higher validation accuracy of 92.51% with 1536 features and very low loss. The best model achieves a verification accuracy of 92.51% and is trained in 52.83 s, thus demonstrating its excellent performance on multi-level classification. The model performance for multi-level classification is summarized in the confusion matrix and the ROC curve in Fig. 3.

### Fourth experiment

In Table 11, the outcomes of our proposed model utilizing PCA and SVD-based feature reduction techniques for multi-class classification are presented. The results of the evaluation show that PCA and SVD are not performing well when these were applied to fused ResNet18 and GoogleNet models on multi-class classification. The accuracy rates that fall within 50.749% to 58.244% imply difficulties in the ability of the fused model to adequately capture and represent complex features as accurately as possible. The

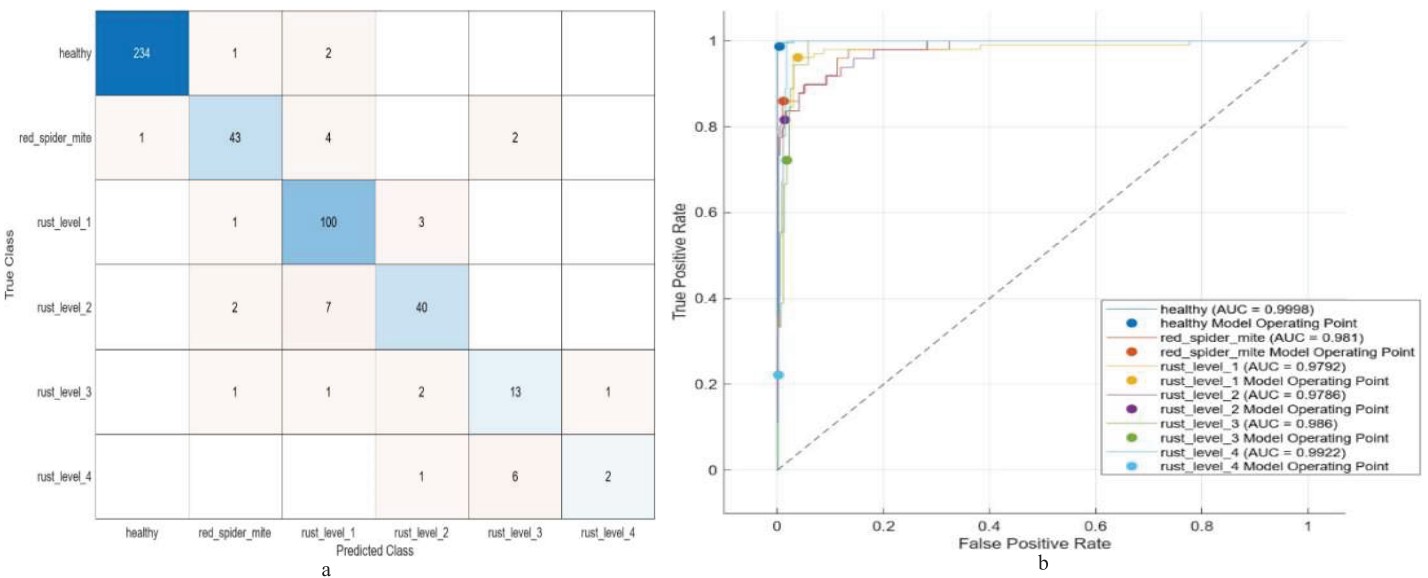

a

b

**Figure 3** (A) Confusion matrix and (B) ROC of the proposed model for multi-level classification.

**Table 11** Results of proposed model with PCA and SVD based feature reduction methods for multi class classification.

|  | SVM | ACC % | ER | Train time | Test time | Healthy | RSM | RL1 | RL2 | RL3 | RL4 |
|---|---|---|---|---|---|---|---|---|---|---|---|
| PCA based proposed | Linear | 50.964 | 49.036 | s | 0.081 | 1 | 0 | 0 | 0 | 0 | 0 |
|  | Quadratic | 58.03 | 41.97 | 0.499 | 0.164 | 0.97 | 0 | 0.33 | 0.14 | 0 | 0 |
|  | Cubic | 57.388 | 42.612 | 0.735 | 0.185 | 0.996 | 0 | 0.252 | 0.12 | 0 | 0 |
|  | Fine Gaussian | 50.964 | 49.036 | 0.532 | 0.183 | 1 | 0 | 0 | 0 | 0 | 0 |
|  | Medium Gaussian | 50.964 | 49.036 | 0.547 | 0.192 | 1 | 0 | 0 | 0 | 0 | 0 |
|  | Coarse Gaussian | 50.964 | 49.036 | 2.008 | 0.434 | 1 | 0 | 0 | 0 | 0 | 0 |
| SVD based proposed | Linear | 50.749 | 49.251 | 0.768 | 0.097 | 1 | 0 | 0 | 0 | 0 | 0 |
|  | Quadratic | 58.244 | 41.756 | 0.789 | 0.373 | 0.996 | 0 | 0.311 | 0.08 | 0 | 0 |
|  | Cubic | 55.889 | 44.111 | 0.882 | 0.412 | 0.996 | 0 | 0.223 | 0.04 | 0 | 0 |
|  | Fine Gaussian | 54.176 | 45.824 | 0.837 | 0.21 | 1 | 0 | 0 | 0 | 0 | 0 |
|  | Medium Gaussian | 50.749 | 49.251 | 0.751 | 0.395 | 1 | 0 | 0 | 0 | 0 | 0 |
|  | Coarse Gaussian | 50.749 | 49.251 | 0.969 | 0.424 | 1 | 0 | 0 | 0 | 0 | 0 |

performance of dimensionality reduction techniques, such as PCA and SVD, depends on data peculiarities in addition to classification task complexity. For fused models that have complicated features, it may be necessary to use alternative methods or continue training to achieve better results in terms of enhancing accuracy of classification.

### Fifth experiment

The hybrid model of GoogleNet and Resnet-18, coupled with ANOVA and chi-square feature selection techniques, had a desirable improvement in performance under multi-level classification as observed above. The results show the ANOVA model achieved an accuracy of 90.15% and an error rate were 9.85%, demonstrating efficiency with

**Table 12 Results of proposed model with Chi-square and ANOVA based feature selection methods for multi-class classification.**

| | SVM | ACC % | ER | Training time | Testing time | Healthy | RSM | RL1 | RL2 | RL3 | RL4 |
|---|---|---|---|---|---|---|---|---|---|---|---|
| ANOVA based proposed | Linear | 90.15 | 9.85 | 0.271 | 0.047 | 0.987 | 0.82 | 0.893 | 0.88 | 0.474 | 0.111 |
| | Quadratic | 90.578 | 9.422 | 0.24 | 0.092 | 0.992 | 0.8 | 0.874 | 0.88 | 0.526 | 0.444 |
| | Cubic | 90.578 | 9.422 | 0.257 | 0.113 | 0.992 | 0.8 | 0.883 | 0.88 | 0.526 | 0.333 |
| | Fine Gaussian | 50.749 | 49.251 | 0.307 | 0.141 | 1 | 0 | 0 | 0 | 0 | 0 |
| | Medium Gaussian | 88.009 | 11.991 | 0.217 | 0.099 | 0.987 | 0.82 | 0.903 | 0.8 | 0.158 | 0 |
| | Coarse Gaussian | 72.805 | 27.195 | 0.244 | 0.108 | 0.987 | 0 | 1 | 0.06 | 0 | 0 |
| Chi-square based proposed | Linear | 90.15 | 9.85 | 0.252 | 0.041 | 0.983 | 0.82 | 0.932 | 0.8 | 0.526 | 0.111 |
| | Quadratic | 91.863 | 8.137 | 0.242 | 0.064 | 0.983 | 0.86 | 0.922 | 0.8 | 0.684 | 0.556 |
| | Cubic | 91.863 | 8.137 | 0.244 | 0.072 | 0.983 | 0.88 | 0.932 | 0.78 | 0.632 | 0.556 |
| | Fine Gaussian | 51.178 | 48.822 | 0.302 | 0.1 | 1 | 0 | 0.019 | 0 | 0 | 0 |
| | Medium Gaussian | 89.293 | 10.707 | 0.286 | 0.069 | 0.983 | 0.82 | 0.951 | 0.82 | 0.211 | 0 |
| | Coarse Gaussian | 71.306 | 28.694 | 0.236 | 0.076 | 0.975 | 0 | 0.99 | 0 | 0 | 0 |

training and testing times of 0.271 and 0.047 s, respectively. Among all models tested, the chi-square-based model stood out as the top performer by achieving an accuracy of 91.86% and an error rate of 8.13%; it has also shown rather efficient training time, which lasted for only 0.242 s, whereas testing took a bit longer, yet remained almost sufficient, 0.064 s.

By reducing the number of features from 1,536 to 500, the feature selection component of the ANOVA-based model highlights how effective feature selection is at improving the model's performance. Table 12 presents the performance of the suggested model using feature selection techniques based on ANOVA and chi-square for multi-class classification. In the context of multi-level classification, Fig. 4 also shows the confusion matrix for the suggested model's ANOVA-based quadratic SVM and chi-square-based cubic SVM.

Table 13 shows the strong indicator of improvement achieved by the suggested method over the current models. Notwithstanding these findings, the model's encouraging results in identifying leaf diseases, particularly in the early stages, demonstrate its usefulness. Feature selection methods combined with convolutional neural networks work well to maximize yield and enable prompt treatment.

## DISCUSSION

The hybrid CNN–SVM approach introduced here was always performing at high classification levels for a range of configurations and datasets, compared to previous studies (*Binney & Ren, 2022*; *Sorte et al., 2019*; *Novtahaning, Shah & Kang, 2022*; *Esgario, Krohling & Ventura, 2020*; *Ayikpa et al., 2022*). The best configuration (GoogLeNet–ResNet18) feature fusion using PCA (98% variance) and subsequent Gaussian RBF SVM classification recorded the best accuracy level of 99.78% on the BRACOL dataset. Besides accuracy, the model also achieved precision: 99.6%, Recall: 99.74%, and F1-score: 99.8%, reflecting an overall balanced performance despite class imbalance. All classes had AUC-ROC higher than 0.99, which implies there was great separability.

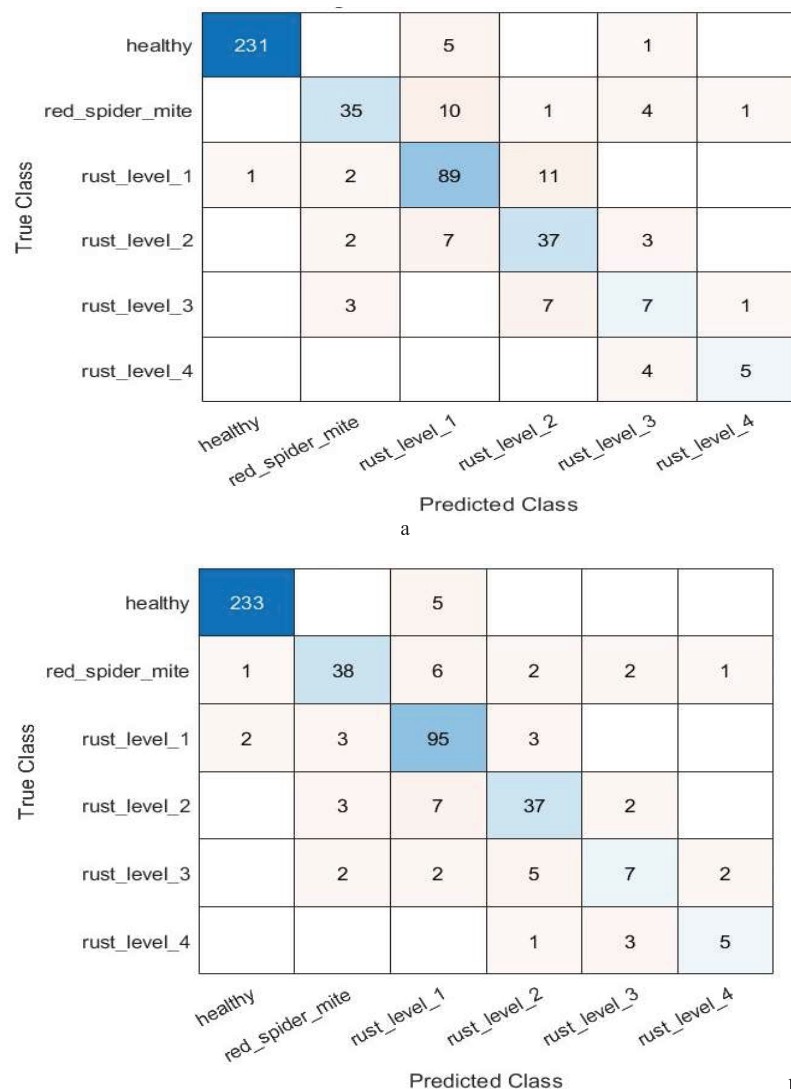

**Figure 4 Confusion matrix of the (A) ANOVA-based quadratic SVM. (B) Chi-square-based cubic SVM of the proposed model for multi-level classification.**

**Table 13 Comparison of the proposed method with previous methods.**

| Model | Accuracy | F1-score | Precision | Recall |
|---|---|---|---|---|
| *Gheorghiu et al. (2024)* | 78.2 | 63.4 | 66.1 | 64.3 |
| *Faisal, Leu & Darmawan (2023)* | 84.29 | 83.64 | 84.67 | 84.29 |
| *Fenu & Malloci (2022)* | 79.83 | – | – | – |
| Proposed method | 91.863 | 90.77 | 89.6 | 90.8 |

Confusion matrices for all the multi-class experiments show that misclassifications between visually equivalent disease categories, such as early and late blight, occurred most frequently. This per-class detailed analysis confirms the robustness of the model over the majority classes and minority classes.

For feature extraction, the evaluation used six Gaussian kernels: coarse, cubic, fine, linear, medium, and quadratic; GoogLeNet and ResNet18 outperformed the others. Compared to other classifiers, the accuracy of the proposed method was 99.78% for binary classification and 91.86% for multi-class classification. While ANOVA feature selection performed best overall, in multi-class classification tasks, chi-square selection showed slightly better performance. This suggests that different feature selection techniques may be more suitable for specific classification scenarios.

The proposed framework achieves enhanced performance by combining complementary CNN architectures (GoogLeNet and ResNet18) with optimized SVM classification. The framework leverages the strength of both deep learning and conventional machine learning, enabling effective and efficient feature representation, enhanced generalization, and reduced computational cost compared to CNN-alone or handcrafted feature-based methods.

## CONCLUSION

Given the significance of coffee to the global economy and its vulnerability to disease, the development of an automated system for classifying coffee plant diseases is essential. We assessed the effectiveness of well-known pre-trained CNN models (such as EfficientNet, GoogLeNet, ResNet-18, ResNet-50, and ShuffleNet) in identifying critical diseases like Red Spider Mite and different stages of coffee rust as well as in classifying healthy coffee plants.

The study used advanced CNN architecture to achieve state-of-the-art accuracy and optimize computational efficiency. GoogLeNet/ResNet18 was the best feature extraction when using six Gaussian kernels. The approach achieved 99.78% and 91.86% accuracy for binary and multi-class classification, respectively. ANOVA was mostly the best choice for feature selection, but chi-square performed slightly better for multi-class issues.

These findings underscore the huge potential of CNN-based models to improve the accuracy and reliability of diagnosing diseases in coffee plants. The application of these advanced technologies can significantly improve the quality of products, create consumer confidence, and raise the bar of industry standards. Further, big data analytics and digital agriculture are set to kick off the transformation of smart coffee growing and provide an avenue for higher quality outputs at lower costs with increased business volume. This transformation, expected to boost industry profits, is also set to nurture sustainable development in the global coffee business.

The development of farmer-friendly interfaces should therefore be given top priority in all future research to make these tools usable and accessible. Research into where agricultural experts will be involved will be vital in determining the field applicability of the models. Likewise, the environmental impact assessment results will greatly help the sustainability of these technologies for coffee production. These projects will collaborate in sustaining, increasing the resilience, and viability of real-world coffee plant disease classification systems. Our study's limitations include the model's high computational requirements, which make it less practical for real-time applications on devices with

limited resources. Additionally, environmental factors may cause variations in model performance across different coffee-growing regions.

### Funding

This work was supported by Princess Nourah bint Abdulrahman University Researchers Supporting Project number (PURSP2025R893), Princess Nourah bint Abdulrahman University, Riyadh, Saudi Arabia. The funders had no role in study design, data collection and analysis, decision to publish, or preparation of the manuscript.

### Grant Disclosures

The following grant information was disclosed by the authors:
Princess Nourah bint Abdulrahman University Researchers Riyadh, Saudi Arabia: PURSP2025R893.

### Competing Interests

The authors declare that they have no competing interests.

### Author Contributions

- Hanin Ardah analyzed the data, authored or reviewed drafts of the article, and approved the final draft.
- Maher Alrahhal conceived and designed the experiments, performed the experiments, analyzed the data, performed the computation work, prepared figures and/or tables, authored or reviewed drafts of the article, and approved the final draft.
- Walaa M. Abd-Elhafiez conceived and designed the experiments, performed the experiments, analyzed the data, performed the computation work, prepared figures and/ or tables, authored or reviewed drafts of the article, and approved the final draft.
- Doaa Trabay conceived and designed the experiments, performed the experiments, analyzed the data, performed the computation work, prepared figures and/or tables, authored or reviewed drafts of the article, and approved the final draft.

### Data Availability

The BRACOL dataset is available at Mendeley: Parraga-Alava, Jorge; Cusme, Kevin; Loor, Angélica; Santander, Esneider (2019), "RoCoLe: A robusta coffee leaf images dataset", Mendeley Data, V2, doi: 10.17632/c5yvn32dzg.2.

The code is available at GitHub and Zenodo:
- https://github.com/DrMaherAlrahhal/coffe-code.
- w. (2025). coffee code. Zenodo. https://doi.org/10.5281/zenodo.17470672.

### Supplemental Information

Supplemental information for this article can be found online at http://dx.doi.org/10.7717/peerj-cs.3386#supplemental-information.

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
