# Peer review of "Robust coffee plant disease classification using deep learning and advanced feature engineering techniques"

_PeerJ Computer Science, doi:10.7717/peerj-cs.3386_

## Round 0.1 · original submission · Major Revisions

· Academic Editor

Major Revisions

**Language Note:** When you prepare your next revision, please either (i) have a colleague who is proficient in English and familiar with the subject matter review your manuscript, or (ii) contact a professional editing service to review your manuscript. PeerJ can provide language editing services - you can contact us at [email protected] for pricing (be sure to provide your manuscript number and title). – PeerJ Staff

·

Basic reporting

1. The manuscript is generally well-written, clear, and logically structured. Figures, tables, and references are relevant and appropriately cited.

2. The abstract and introduction clearly state the research objectives. Minor improvements in the explanation of methods (particularly CNN selection and feature engineering steps) would enhance clarity and reproducibility.

Experimental design

1. Please include specifics about training configurations, such as learning rate, optimizer, batch size, number of epochs, and whether data augmentation was applied.

2. Clarify whether cross-validation or a fixed train-test split was used. If the latter, discuss justification and potential limitations.

3. Feature Engineering Parameters: Indicate how the number of components and thresholds were selected.

Validity of the findings

1. Include precision, recall, and F1-score to provide a more balanced evaluation, especially in the case of class imbalance.

2. If possible, add statistical significance testing or confidence intervals to support claims of superiority over other models.

3. Briefly discuss the model’s generalizability—e.g., whether it was tested on unseen or real-world data beyond the BRACOL dataset.

Additional comments

1. This is a well-executed study with practical significance for agriculture and precision farming. The integration of deep learning and feature engineering is commendable.

2. Minor improvements in methodology explanation and evaluation metrics would enhance clarity and impact.

3. Overall, the work makes a valuable contribution to the field and is suitable for publication after addressing the suggested minor revisions.

Reviewer 2 ·

Basic reporting

No new methods were proposed.

Experimental design

This study used existing methods which lack novelty in the field of research.

Validity of the findings

Results are presented. However, novelty and contribution are absent.

Reviewer 3 ·

Basic reporting

Strengths:
- The manuscript is written in clear and mostly professional English. It communicates the research objectives and results effectively.

- The Introduction provides adequate context and references a wide range of related studies.

- The structure aligns with PeerJ standards: abstract, introduction, related work, methods, results, and discussion are all present and well organized.

- Figures, tables, and references are appropriately used and cited.

Suggestions for improvement:
- Some sentences are overly verbose or awkwardly phrased. A thorough English language polish is recommended.

- The motivation in the abstract and introduction could be more sharply defined, especially with regard to what specific gap this study fills.

- Data augmentation techniques are briefly mentioned, but could benefit from more specifics.

- Consider condensing the detailed SVM kernel math into an appendix to maintain flow for general readers.

Experimental design

Strengths:

- The experimental design is comprehensive, involving multiple CNN architectures, SVM kernels, and combinations thereof.

- Preprocessing, feature extraction, feature selection, and dimensionality reduction steps are clearly detailed.

- The BRACOL dataset is public and available; code is provided via GitHub, ensuring reproducibility.

- Multiple metrics (accuracy, F1-score, AUC-ROC, etc.) are used for thorough model evaluation.

Suggestions for improvement:

- The rationale for combining CNN models could be better justified in terms of complementary feature representations.

- The choice of PCA and SVD thresholds for variance retention should be explicitly reported.

- While the SVM kernels are well explained, the selection criteria or grid search strategy for tuning hyperparameters is not detailed.

Validity of the findings

Strengths:

- The authors critically evaluate different feature selection and reduction techniques (ANOVA, Chi-square, PCA, SVD).

- Model limitations and future directions are acknowledged, such as computational requirements and the need for real-time applications.

Suggestions for improvement:

- Results on multi-class classification are strong, but the drop in performance after dimensionality reduction warrants deeper discussion.

- Some of the results (e.g., 99.78% accuracy) are unusually high. While they are consistent with reported methodology, overfitting should be explicitly ruled out (e.g., via k-fold CV or unseen test sets).

- It would be beneficial to include a confusion matrix for all multi-class experiments (not just the final one).

---

## Round 0.2 · accepted · Accept

· Academic Editor

Accept

The reviewers seem satisfied with the latest corrections and therefore I can recommend this article for acceptance.

·

Basic reporting

The manuscript is clearly written in professional English with proper structure and flow. The introduction effectively explains the motivation and context of the study, and the literature review is relevant and well-referenced. Minor improvement could be made by briefly including recent transformer-based models for completeness. Overall, it meets the standards for clarity, context, and technical accuracy.

Experimental design

The study aligns well with the journal’s scope and demonstrates a rigorous experimental approach. The methodology is well-structured, with clear descriptions of the CNN architectures (GoogLeNet and ResNet18), optimization parameters, and evaluation metrics. However, additional details on dataset composition, data preprocessing steps, and train-test split ratio would improve reproducibility. Including brief notes on computing infrastructure and runtime efficiency is also recommended. Overall, the experimental design is technically sound, ethically appropriate, and supports the study’s objectives effectively.

Validity of the findings

The experiments are well-executed, and the evaluation metrics strongly support the reported results. The conclusions are clearly stated and consistent with the objectives outlined in the introduction. The proposed hybrid model demonstrates both novelty and practical value in coffee disease classification. However, the paper could be further strengthened by including a brief discussion of limitations (e.g., dataset diversity or scalability) and future research directions. Overall, the findings are valid, well-supported, and contribute meaningfully to the field.

Additional comments

The paper presents a strong contribution to AI-based agricultural research, particularly in plant disease detection using hybrid deep learning models. The integration of CNNs with statistical feature engineering techniques is innovative and well-executed. With minor additions—such as more dataset details, computational efficiency analysis, and a short note on future extensions—the manuscript will be even more impactful. Overall, it is a well-prepared and publication-worthy work.

Reviewer 3 ·

Basic reporting

-

Experimental design

-

Validity of the findings

-

Additional comments

The authors have addressed my comments.